# Circulating Cell-Free DNA-Based Liquid Biopsy Markers for the Non-Invasive Prognosis and Monitoring of Metastatic Pancreatic Cancer

**DOI:** 10.3390/cancers12071754

**Published:** 2020-07-01

**Authors:** Marta Toledano-Fonseca, M. Teresa Cano, Elizabeth Inga, Rosa Rodríguez-Alonso, M. Auxiliadora Gómez-España, Silvia Guil-Luna, Rafael Mena-Osuna, Juan R. de la Haba-Rodríguez, Antonio Rodríguez-Ariza, Enrique Aranda

**Affiliations:** 1Maimónides Biomedical Research Institute of Córdoba (IMIBIC), E14004 Córdoba, Spain; martatoledanofonseca@gmail.com (M.T.-F.); v22gulus@uco.es (S.G.-L.); b72meosr@uco.es (R.M.-O.); juahaba@gmail.com (J.R.d.l.H.-R.); earandaa@seom.org (E.A.); 2Cancer Network Biomedical Research Center (CIBERONC), E28029 Madrid, Spain; auxiliadora.gomez.sspa@juntadeandalucia.es; 3Medical Oncology Department, Reina Sofía University Hospital, E14004 Córdoba, Spain; maytecano79@gmail.com (M.T.C.); e.ingasaavedra@gmail.com (E.I.); rosarodriguezalonso@gmail.com (R.R.-A.); 4Department of Medicine, Faculty of Medicine, University of Córdoba, E14004 Córdoba, Spain

**Keywords:** cell-free DNA, liquid biopsy, MAF, pancreatic cancer, RAS mutation

## Abstract

Liquid biopsy may assist in the management of cancer patients, which can be particularly applicable in pancreatic ductal adenocarcinoma (PDAC). In this study, we investigated the utility of circulating cell-free DNA (cfDNA)-based markers as prognostic tools in metastatic PDAC. Plasma was obtained from 61 metastatic PDAC patients, and cfDNA levels and fragmentation were determined. BEAMing technique was used for quantitative determination of RAS mutation allele fraction (MAF) in cfDNA. We found that the prognosis was more accurately predicted by RAS mutation detection in plasma than in tissue. RAS mutation status in plasma was a strong independent prognostic factor for both overall survival (OS) and progression-free survival (PFS). Moreover, RAS MAF in cfDNA was also an independent risk factor for poor OS, and was strongly associated with primary tumours in the body/tail of the pancreas and liver metastases. Higher cfDNA levels and fragmentation were also associated with poorer OS and shorter PFS, body/tail tumors, and hepatic metastases, whereas cfDNA fragmentation positively correlated with RAS MAF. Remarkably, the combination of CA19-9 with MAF, cfDNA levels and fragmentation improved the prognostic stratification of patients. Furthermore, dynamics of RAS MAF better correlated with patients’ outcome than standard CA19-9 marker. In conclusion, our study supports the use of cfDNA-based liquid biopsy markers as clinical tools for the non-invasive prognosis and monitoring of metastatic PDAC patients.

## 1. Introduction

Pancreatic cancer is the fourth leading cause of cancer death in Europe in both males and females, with the lowest survival rate of all cancers and responsible for over 95,000 deaths every year [1,2]. While, the death rates of the most common cancers have mostly declined over the past decades, the mortality rate of pancreatic cancer remains flat or slightly increases over time [3]. Poor prognosis is associated with diagnosis at advanced stage, due to a lack of detection methods, as well as resistance to therapy. Pancreatic ductal adenocarcinoma (PDAC) represents more than 90% of all pancreatic cancer and the vast majority of deaths are associated with this tumor type. Approximately 60–70% of patients have a primary tumor, located in the head of pancreas, while 20% and 25% are located in the body, and tail, respectively. Moreover, PDAC metastasizes mainly to liver, abdomen and lungs [4].

KRAS mutation is the initiating genetic event for PDAC, and this oncogene is mutationally activated in 94% of pancreatic ductal tumors [5]. KRAS mutational status has been usually analyzed in tumor tissue, but obtaining biopsy specimens from pancreatic lesions may be difficult and requires invasive procedures, such as endoscopic ultrasound-guided fine needle aspiration (EUS-FNA). However, EUS-FNA provides a limited number of cells for molecular profiling and the high stromal content of pancreatic tumors impairs its efficacy for PDAC diagnosis [6]. Moreover, tumor tissue is only available at diagnosis, but not at different time points of the disease to monitor tumor burden during treatment. In this scenario, circulating cell-free DNA (cfDNA)-based liquid biopsy represents a promising non-invasive tool for the diagnosis, prognosis and management of PDAC patients [7]. Hence, the analysis of KRAS mutations in cfDNA has been proposed as non-invasive surrogate for tissue biopsies in patients with pancreatic cancer [8]. On the other hand, cfDNA levels have been shown to be prognostic for clinical outcome in metastatic cancer [9,10].

Therefore, the present study aimed to evaluate the different cfDNA-based liquid biopsy markers as prognostic tools for the management care of metastatic PDAC patients.

## 2. Results

### 2.1. Clinicopathological Characteristics and RAS Mutation Analysis from Plasma and Tissue

Sixty-one patients were included in the study between May 2017 and December 2019. Thirty-four patients were men and 27 were women, and they ranged in age from 40 to 84 years, with a median of 65 years of age (baseline characteristics are summarized in Table 1). All patients had pancreatic ductal adenocarcinoma (PDAC) with distant metastases at diagnosis, and the most frequent site of metastasis was the liver (78.7%). Primary tumor was localized in tail, body and head of pancreas in 27.9%, 41%, and 29.5% of patients, respectively. Most patients (78.7%) had a good baseline ECOG performance status (PS), and a majority (75.4%) received first line gemcitabine-based regimes. For the analysis of RAS mutational status, primary tumor tissue was available in 70.5% (43/61) of patients. Whereas, basal blood samples were obtained from all patients before any treatment. RAS mutation was detected in 76.7% (33/43) of tissue samples and in 77% (47/61) of basal plasma samples. The percentage of patients with RAS mutation was comparable to other cohort studies [11,12,13]. Mutation in codon 12 of the KRAS gene was found in 93.6% (44/47), and 93.9% (31/33) of plasma and tissue samples, respectively (Appendix A). The overall concordance between plasma and tissue RAS analysis was 79.1%.

### 2.2. Detection of RAS Mutations in cfDNA Predicts Poor Prognosis in Metastatic PDAC Patients

The presence of RAS mutations in plasma cfDNA was analysed in 61 metastatic PDAC patients. Detection of RAS mutation in plasma was associated with shorter patient overall survival (OS) (169 versus 372.5 days; *p* = 0.0004; Table 2, Figure 1A). Besides, prognosis was more accurately predicted by RAS mutation analysis in cfDNA than by tissue analysis (43 patients, RAS mutation in tissue: log-rank *p* = 0.0730; RAS mutation in cfDNA: *p* = 0.0068; Table 2, Figure 1B,C). RAS mutation detection in cfDNA was also a predictive factor of poor progression-free survival (PFS) in metastatic PDAC patients (93.5 versus 313.5 days; *p* < 0.0001; Table 3, Figure 2A). Likewise, tissue analysis was a worse predictive factor of PFS than cfDNA (RAS mutation in tissue: *p* = 0.0172; RAS mutation in cfDNA: *p* = 0.0019; Table 3, Figure 2B,C). Finally, multivariate analysis revealed that KRAS mutation status in plasma was a strong independent prognostic factor for both OS (HR 5.692, 95% CI 1.497–21.636; *p* = 0.011) and PFS (HR 8.631, 95% CI 2.311–32.236; *p* = 0.001) (Table 4).

### 2.3. Higher RAS Mutational Load in cfDNA is Associated with Poor Prognosis in Metastatic PDAC

For the 47 patients with detectable plasma RAS mutations, the median mutation allele fraction (MAF) was 2.92% (range 0.02–29.33%). As shown in Figure 3 a higher RAS mutational load in cfDNA was associated with poor OS (142 versus 310 days; *p* = 0.0261; cut-off value: 0.351%, with 82.5% sensitivity and 100% specificity; Table 2, Figure 3A) and poor PFS (85 versus 175 days; *p* = 0.0556; cut-off-value: 0.351%, with 83% sensitivity and 48% specificity; Table 3, Figure 3B). Moreover, multivariate analysis identified MAF in cfDNA as an independent risk factor for poor OS (HR 1.070, 95% CI 1.001–1.143; *p* = 0.047) (Table 4). Although, no differences were observed in the MAF values according to the number of metastatic lesions, higher MAF values were strongly associated with primary tumors located in the body/tail of the pancreas (*p* = 0.0281, Figure 4A) and liver metastases (*p* = 0.0072, Figure 4B). In this regard, the primary tumor location (OS *p* = 0.5802; PFS *p* = 0.5318) or the number of metastatic lesions (OS *p* = 0.3380; PFS *p* = 0.6304) were not related to OS or PFS. Whereas, significant poorer OS and PFS were observed in patients with hepatic lesions compared to patients with metastasis affecting other organs (OS 157 versus 339 days; *p* = 0.0114; PFS 86 versus 272; *p* = 0.0048) (Table 2; Table 3).

### 2.4. Higher cfDNA Concentration and Fragmentation Levels Are Associated with Poorer Survival in Metastatic PDAC Patients

The median cfDNA concentration in plasma of PDAC patients was 33 ng/mL (range 10–700), while the fragment size of plasma cfDNA ranged between 100–1100 bp, with a prominent mode at 135 pb for the shortest fragments detected. In this study cfDNA fragmentation was defined as the percentage of shortest fragments to total cfDNA. As shown in Figure 5, cfDNA concentration was significantly higher in those patients in whom plasma RAS mutations were detected (42.65 versus 24.71 ng/mL, *p* = 0.0057; Figure 5A). Although not significant, higher cfDNA fragmentation was observed in RAS mutated patients (Figure 5B), and a significant positive correlation between cfDNA fragmentation and KRAS MAF was found (r = 0.31, *p* = 0.0189).

When metastatic PDAC patients were stratified according to plasma cfDNA concentration, those with higher values (>26.46ng/mL) had a poorer OS rate (149.5 versus 285 days, *p* = 0.0057; cut-off value: 26.46 ng/mL, with 70.6% sensitivity and 64.1% specificity; Figure 5C, Table 2). Also, higher plasma cfDNA concentration was associated with shorter PFS (86.5 versus 149.5 days, *p* = 0.0107, cut-off value: 26.46 ng/mL, with 67.7% sensitivity and 100% specificity; Figure 5D, Table 3). Similarly, a higher percentage of plasma cfDNA fragmentation in metastatic PDAC patients was significantly associated with a poorer OS (116 versus 197 days, *p* = 0.0297; cut-off value: 38.08%, with 27.2% sensitivity and 100% specificity; Figure 5E, Table 2) and PFS rates (145 versus 81 days, *p* = 0.0101; cut-off value: 38.08%, with 27.2% sensitivity and 100% specificity; Figure 5F, Table 3).

Plasma cfDNA concentration or fragmentation were not associated with number of metastatic lesions (*p* = 0.4928; *p* = 0.7735). However, there was an association between cfDNA fragmentation and primary tumor located in body/tail compared to head of the pancreas (15.27 versus 9.27%, *p* = 0.0401) (Figure 6A) and a trend towards higher cfDNA concentration in the plasma of metastatic PDAC patients with body/tail tumors (36.17ng/mL) compared with those with tumors in the head of the pancreas (26.23ng/mL, *p* = 0.0691) (Figure 6B). Also, patients with hepatic metastasis displayed higher cfDNA levels in plasma (38.10ng/mL), when compared with those patients with other metastatic locations (28.93ng/mL, *p* = 0.0547) (Figure 6D). Similarly, a trend towards higher cfDNA fragmentation was observed in patients with metastatic lesions in the liver (12.165%), compared with those with metastases elsewhere (10.655%, *p* = 0.3257) (Figure 6C).

### 2.5. Multiparameter Liquid Biopsy Refines Prognostic Stratification of Metastatic PDAC Patients

In our cohort, CA19-9 demonstrated some prognostic value, with higher baseline levels associated with poorer OS and PFS rates (OS 125 versus 202.5 days, *p* = 0.0408; cut-off value: 45,500 U/mL, with 16.2% sensitivity and 80.9% specificity; PFS 72 versus 143 days, *p* = 0.0289; cut-off value: 45,500 U/mL, with 23% sensitivity and 93.7% specificity; Table 2; Table 3). No association was found between CA19-9 levels and RAS mutation status (*p* = 0.2909), primary tumor location (*p* = 0.5053), number of metastasis (*p* = 0.4723), location of metastatic lesions (*p* = 0.4908), MAF (*p* = 0.1642), cfDNA levels (*p* = 0.7692) or cfDNA fragmentation (*p* = 0.2769).

Remarkably, the combination of CA19-9 with liquid biopsy improved the prognostic stratification of metastatic PDAC patients. A scoring system was applied by combining CA19-9 with MAF, cfDNA concentration and cfDNA fragmentation. Positive or negative values were assigned depending on whether the corresponding marker was above (positive) or below (negative) the cut-off with prognostic value in OS. Thus, score 0 was defined as negative for all markers; score 1 was defined as positive for 1 marker; and score 2 was defined as positive for 2, 3 or 4 markers. As shown in Figure 7, those patients with score 2 displayed poorer survival outcomes in comparison with those patients with score 0 and score 1 in Kaplan-Meier analysis (*p* = 0.0002, and *p* = 0.0072, respectively).

### 2.6. RAS Mutational Load in cfDNA Enables Monitoring of Disease Progression and Response to Therapy in Metastatic PDAC Patients

Due to the limitations in CA19-9 as a reliable marker of pancreatic cancer, the utility of circulating MAF was compared to CA19-9 in monitoring disease progression and response to therapy in metastatic PDAC patients. No RAS mutation was detected in blood at baseline in two of the seven monitored patients, but it was detected in disease progression. In patient 1, KRAS codon 12 mutation was found in tissue but not in blood at baseline. Eventually, a novel NRAS mutation was detected during stable disease and a circulating KRAS codon 12 mutation was detected later in blood, along with both elevation of CA19-9 levels and disease progression revealed by radiological criteria and followed by rapid deterioration and death (Figure 8A). In patient 2, no RAS mutation was detected at baseline in either tissue or blood, but a KRAS codon 12 mutation was detected later in blood at progression of the disease (Figure 8B).

In the three patients (3, 4 and 5) in whom RAS mutation was detected at baseline in blood, circulating MAF dropped following treatment and concurring with lower CA19-9 levels and partial response (PR) to therapy (Figure 8C–E). In patient 3, circulating KRAS mutation level markedly declined at PR and rose again at disease progression, along with the detection of a novel circulating NRAS mutation (Figure 8C). In patient 4, KRAS mutation remained undetectable in blood, while CA9-19 levels were low and the disease was stable (SD), but unlike CA19-9, MAF was augmented again at the progression of disease (Figure 8D). In patient 5, circulating KRAS mutation dropped to undetectable levels in the stable disease. Despite standard criteria and CA19-9 levels in the following monitoring suggested stable disease, KRAS mutation was detected again in plasma anticipating disease progression (Figure 8E).

Finally, in patients 6 and 7, circulating RAS mutation levels increased during treatment, compared to baseline levels (Figure 8F,G). Notably, the increase in circulating MAF was associated with a very short survival period (5 months since diagnosis) in these patients.

As a whole, the above results suggest that the dynamics of circulating RAS mutation may better correlate with patients’ outcome and survival compared with standard CA19-9 marker. Accordingly, a significant correlation was found between the increase in MAF (*r* = −0.65, *p* = 0.02), but not in CA19-9 (*r* = 0.09, *p* = 0.78) and survival time (Figure 9). Hence, higher increases in circulating RAS mutation during patient monitoring predicted a shorter survival time.

## 3. Discussion

Non-invasive, reliable, and reproducible cfDNA-based liquid biopsy markers may help in the management of cancer patients. This is particularly relevant in the case of PDAC, where the high stromal content makes it difficult to obtain molecular information through cytopathological analysis. However, there is no consensus about the techniques, mutations or type of material in liquid biopsy-based approaches for the prognosis of PDAC patients [11,14,15,16]. In this study we report the utility of cfDNA RAS mutations analysis using the highly sensitive BEAMing technique as prognostic tool for the management care of metastatic PDAC patients.

In agreement with other reports [17], our study supports the value of cfDNA RAS mutations analysis as a prognostic tool in pancreatic cancer. Therefore, our results show that the presence of RAS mutated cfDNA in plasma predicts poor prognosis in metastatic PDAC patients. Moreover, circulating KRAS mutational status was an independent negative prognostic factor of both OS and PFS. In fact, the prognosis was more accurately predicted by RAS mutation analysis in cfDNA than in tissue. The allelic ratio and dosage of mutated KRAS may impact on PDAC biology [18], and KRAS MAF in cfDNA has been found to correlate with clinical stage and outcome in PDAC [13,19]. In this regard, our results reveal that circulating KRAS MAF in cfDNA predicted survival in metastatic PDAC patients. Importantly, in our study, KRAS MAF in cfDNA was an independent negative prognostic factor of OS by multivariate analysis. Recently, KRAS MAF in DNA from circulating exosomes, but not in cfDNA, was found to be an independent prognostic factor of OS in metastatic PDAC patients [19]. However, our study demonstrates that highly sensitive approaches, such as BEAMing, may also reveal the independent prognostic value of KRAS MAF in cfDNA of metastatic PDAC patients. Exhaustive analyses on tissue, including laser capture microdissection, could establish the pure ratio of RAS mutated allele in tumor. However, these types of analyses rely on the availability of biopsy material to be adequately performed, which is not the case for a significant number of PDAC patients, and is identifiably the issue by which a liquid biopsy may effectively address.

Although, KRAS mutations are critical for the initiation of pancreatic ductal carcinogenesis, continued mutant KRAS function and oncogenic dosage are still required to maintain the growth of metastatic PDAC [5,20]. On the other hand, gene expression studies revealed that, compared to head localization, body-tail PDAC are more highly proliferative and aggressive [21,22]. Body/tail location is also associated with poor prognosis in advanced disease [23,24,25]. This may explain the reason why, in our cohort, higher values of KRAS MAF in cfDNA of metastatic PDAC patients were significantly associated with primary tumors located in the body/tail of the pancreas and liver metastases. Moreover, the higher MAF observed in patients with liver metastases may be explained by the larger volume of hepatic lesions than the isolated lung and peritoneal metastases [19].

Previous studies have reported the potential prognostic value of cfDNA levels and fragmentation in metastatic cancer [9,10,26], including metastatic PDAC [27]. In our study, higher plasma cfDNA concentrations were significantly associated with poorer OS and shorter PFS. Patients with hepatic metastasis displayed higher cfDNA levels, compared with those patients with other metastatic locations.

Despite the lack of knowledge about the precise mechanisms of cfDNA release into circulation, the role of apoptosis is becoming clearer [28]. A recent study reported that tumor-derived KRAS mutations in pancreatic cancer are predominantly carried by short and ultra-short cfDNA fragments [29]. This may be the biological explanation for our observation that, in parallel with our KRAS MAF results, a higher cfDNA fragmentation was found in patients with tumors located in the body/tail of the pancreas or with hepatic metastases than other metastatic lesions, likely due to more aggressive tumors. Thus, recent reports showed that body/tail PDAC may have more aggressive tumor biology and higher metastasis rate compared to PDAC in the head which may explain worse clinical outcomes [21,22,23].

CA19-9, also known as sialyl Lewis A antigen, is the currently used biomarker for pancreatic cancer, and several studies have reported the link between CA19-9 levels and survival in metastatic PDAC patients [30,31]. However, CA19-9 have some important limitations, such as false negative results in subjects with Lewis negative genotype and CA19-9 increases in patients with benign pancreatic-biliary diseases [32]. In our cohort, CA19-9 exhibited some prognostic value with higher baseline levels associated with poorer OS and PFS rates. However, our study demonstrates that the combination of CA19-9 with liquid biopsy markers greatly helped in the prognostic stratification of metastatic PDAC patients.

CA19-9 is also used for monitoring treatment response as the reduction of CA19-9 serum levels during treatment are usually associated with longer survival rates. However, in clinical practice, there is no consensus on the interpretation of the change in CA19-9 levels and its role in the management of PDAC patients [33]. Therefore, novel reliable biomarkers are required for monitoring the response of PDAC patients to chemotherapy [34]. In our analysis, the change in circulating KRAS MAF levels was a suitable surrogate marker for monitoring each patient’s response to therapy. Moreover, the rise in MAF levels in some patients was better than CA19-9 in anticipating disease progression, and dynamics of circulating MAF better correlated with patients’ outcome compared with CA19-9. Therefore, our results support MAF as a valuable complementary tool for monitoring the response to chemotherapy treatment in metastatic PDAC patients.

In summary, our study supports cfDNA-based liquid biopsy markers as promising clinical tools for the non-invasive prognosis and monitoring of metastatic PDAC patients.

## 4. Materials and Methods

### 4.1. Patients

Sixty-one patients diagnosed with metastatic PDAC in the Reina Sofía Hospital (Córdoba, Spain) were enrolled in this study from 2017 to 2019. Eligible patients were 18 years or older with histologically confirmed metastatic PDAC and were not treated by chemotherapy or radiotherapy before the enrollment. Metastatic PDAC pathology was confirmed in all patients included in our study by pathological analysis of tumor tissue (*n* = 43) or by cytological analysis (*n* = 18), and by computed tomography. All subjects gave their informed consent for their inclusion in the study. The study was conducted in accordance with the Declaration of Helsinki, and the protocol was approved by the Ethics Committee of Córdoba (Comité de Ética de la Investigación de Córdoba, CEI Córdoba, PANCREAS-BIOPSIA-LIQ protocol, approved on April 26, 2017, Act nº263, ref, 3490). The baseline characteristics of the patients included in the study are listed in Table 1.

### 4.2. Procedures for Sample Analyses

Plasma was obtained from 10 mL of blood collected in Streck cell- free DNA BCT^TM^ tubes before any therapeutic intervention. In seven patients, the plasma was also obtained at specified intervals after the onset of treatment. Blood samples were centrifuged at 1600× *g* during 10 min at room temperature (RT) to separate plasma, followed by centrifugation at 6000× *g* during 10 min at RT to remove any possible remaining cells. Plasma samples were then aliquoted, transferred to cryotubes and stored at −80 °C. QIAamp Circulating Nucleic Acid Kit and the vacuum system QIAvac 24 Plus (Qiagen) were used for cfDNA extraction from 3 mL of plasma and extracted cfDNA was quantified using the Quantus fluorometer (Promega). The High Sensitivity D1000 ScreenTape Assay was used in an Agilent 2200 TapeStation System (Agilent) to analyse cfDNA fragmentation.

OncoBEAM™ RAS assay (Sysmex Inostics GmbH, Baltimore, MD, USA), which detects 34 mutations in KRAS/NRAS codons 12, 13, 59, 61, 117, and 146 was used to analyze RAS mutations in cfDNA and determine MAF in plasma. In brief, OncoBEAM™ RAS Assay started with a conventional PCR to amplify a locus of interest, which included 7 amplicons covering 12 codons and 34 mutations in KRAS/NRAS genes. For each codon a digital PCR was then performed and cfDNA was hybridized with fluorescent probes to quantify by flow cytometry KRAS/NRAS mutant and wild type molecules. This approach allows reliable detection of MAF < 0.1% in cfDNA [35].

In 43 patients, FFPE primary tumor tissue was available for RAS mutation analysis by standard-of-care procedures validated in our hospital. Specifically, the Idylla^TM^ plattform (Biocartis), that utilizes microfluidics processing with specific cartridges and all reagents on board, was employed for RAS mutation analysis in tissue. The process is fully automated, including nucleic acid extraction and, if the results indicate WT KRAS, testing for NRAS mutations is mandatory using another specific cartridge. Serum CA19-9 levels were measured using a standard radioimmunoassay test in the Clinical Laboratory Department of our hospital.

### 4.3. Statistical Analyses

Statistical analyses were performed using SPSS Statistic 20.0.0, GraphPad Prism 7.0 Software and R Software 4.0.0. Overall survival (OS) was calculated from the date of diagnosis to the date of death. Progression-free survival (PFS) was calculated from the start date of therapy until disease progression. The survival rates were estimated using the Kaplan–Meier method and the Log-Rank test was used to identify the prognostic variables. SurvivalROC package in the R software was used to find optimal cut-off values in OS analyzing time-dependent ROC curve. The optimal cut-off value was chosen by minimizing the sum of false negative rate and false positive rate. In each case, the cut-off with prognostic value for OS was also tested for prognosis of PFS. When the optimal cut-off chosen with ROC curves was not able to separate statistically the groups according to the Kaplan-Meier analysis, the R2 Genomics Analysis and Visualization Platform (http://r2.amc.nl) was used to find a cut-off value, using the Kaplan Scan (KaplanScan) feature, based on statistical testing. The Kaplan scanner separates the samples of a dataset into two groups based on values of variable of interest. In the order of values, it uses every increasing value as a cutoff to create 2 groups and test the p-value in a Log-Rank test. Mann-Whitney test was used to compare two groups and ANOVA test for analysis with more than two groups. Multivariate analysis was performed to establish independent prognostic factors using Cox proportional hazards modeling. Graph data are represented as mean ± standard deviation. Correlation analyses were performed using Pearson’s correlation coefficient. All statistical tests were considered significant when *p* < 0.05.

## 5. Conclusions

We evaluated different cfDNA-based liquid biopsy markers as prognostic tools for the management care of metastatic PDAC patients. Our study shows that prognosis was more accurately predicted by RAS mutation analysis in cfDNA than by tissue analysis. Hence, both RAS mutation status and mutational load in cfDNA were independent risk factors for OS. Whereas, a higher cfDNA concentration and fragmentation levels were also associated with poorer survival. Notably, our data support the theory that multi-parameter liquid biopsy may significantly assist in the prognostic stratification of metastatic PDAC patients, while RAS MAF in cfDNA may facilitate with the monitoring of disease progression and response to therapy. Future larger studies with independent cohorts are warranted to validate cfDNA-based liquid biopsy markers for the non-invasive prognosis and monitoring of metastatic PDAC patients.

## Figures and Tables

**Figure 1 cancers-12-01754-f001:**
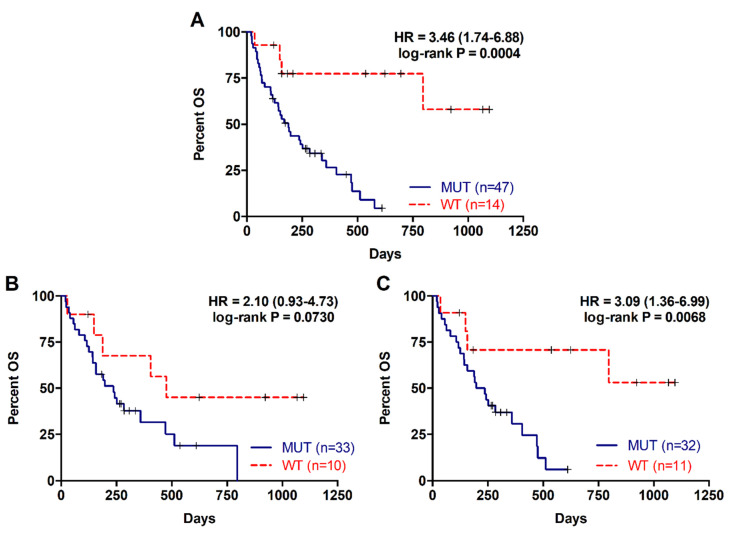
Overall survival rates for patients with metastatic PDAC according to RAS mutation status. (**A**) OS according to RAS mutation status in cfDNA; (**B**) OS according to RAS mutation status in tissue; (**C**) OS according to RAS mutation status in cfDNA of those patients with RAS mutations analyzed in tissue.

**Figure 2 cancers-12-01754-f002:**
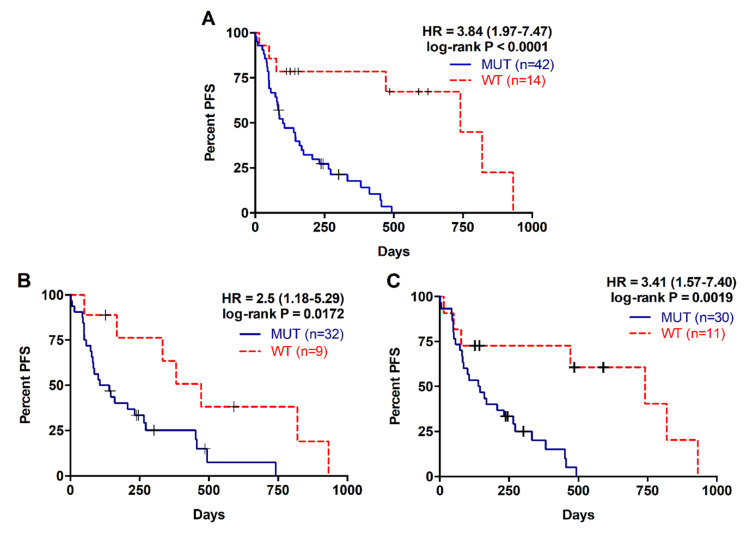
Progression free survival rates for patients with metastatic PDAC according to RAS mutation status. (**A**) PFS according to RAS mutation status in cfDNA; (**B**) PFS according to RAS mutation status in tissue; (**C**) PFS according to RAS mutation status in cfDNA of those patients with RAS mutations analyzed in tissue.

**Figure 3 cancers-12-01754-f003:**
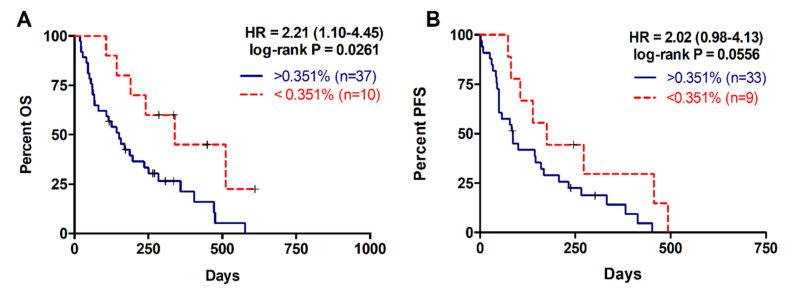
Overall and progression free survival rates according to circulating RAS mutation allele fraction (MAF). (**A**) OS according to circulating MAF (cut-off: 0.351%); (**B**) PFS according to circulating MAF (cut-off: 0.351%).

**Figure 4 cancers-12-01754-f004:**
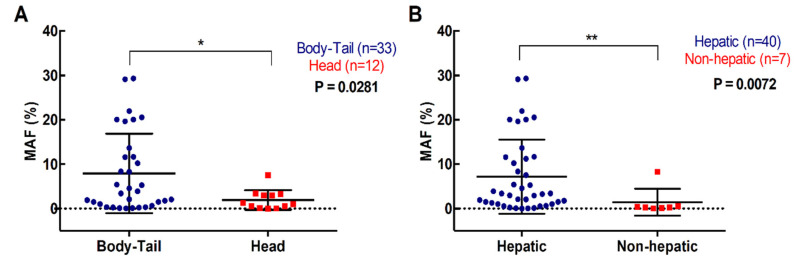
Association of circulating RAS mutation allele fraction (MAF) with primary tumor and metastases location. (**A**) circulating MAF levels in patients with tumor located in the body-tail or the head of the pancreas; (**B**) circulating MAF levels in patients with metastatic lesions in the liver or elsewhere (* *p* < 0.05, ** *p* < 0.01).

**Figure 5 cancers-12-01754-f005:**
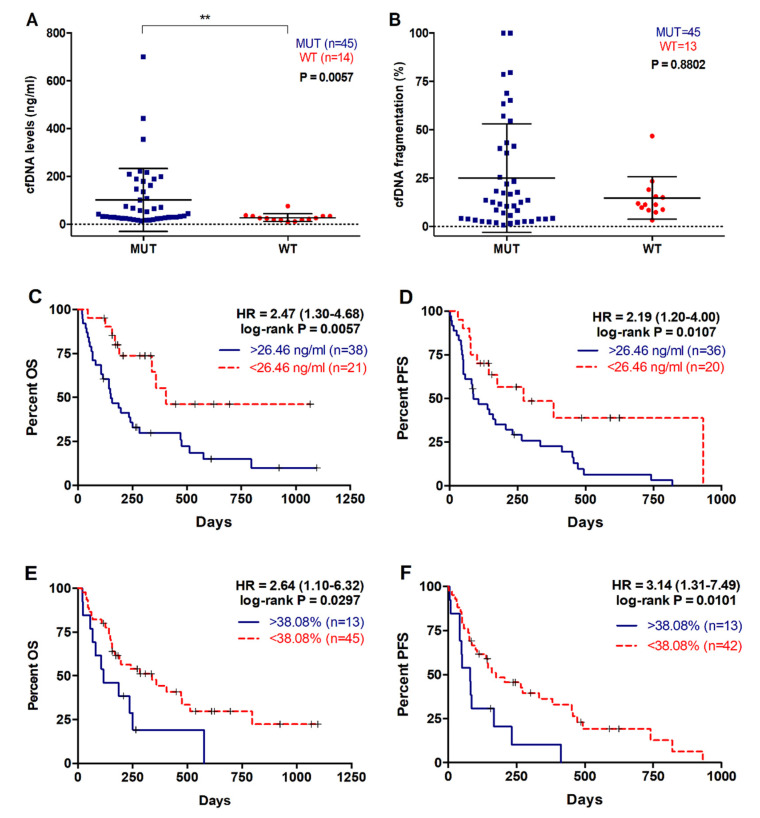
cfDNA concentration and fragmentation in metastatic PDAC patients. (**A**) cfDNA levels and (**B**) fragmentation according to RAS mutational status; (**C**) OS according to cfDNA levels (cut-off: 26.46ng/mL); (**D**) PFS according to cfDNA levels (cut-off: 26.46ng/mL); (**E**) OS according to cfDNA fragmentation (cut-off: 38.08%); (**F**) PFS according to cfDNA fragmentation (cut-off: 38.08%). (** *p* < 0.01).

**Figure 6 cancers-12-01754-f006:**
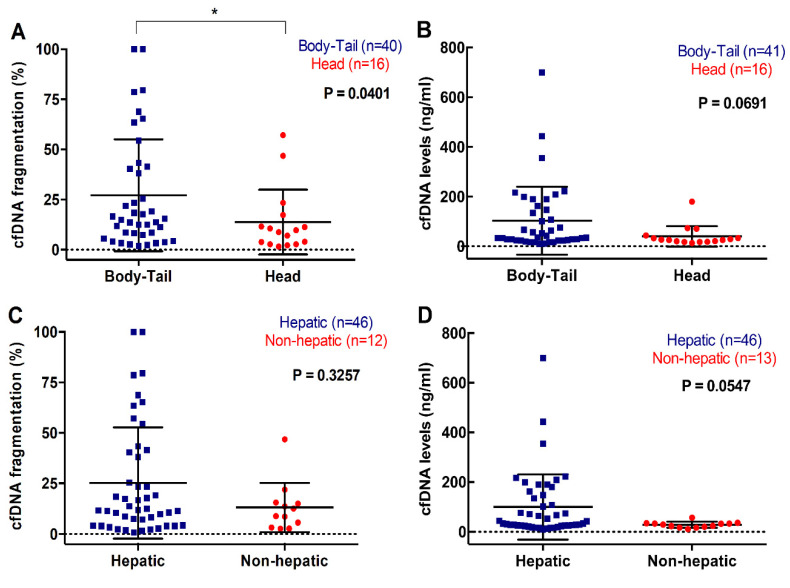
Association between cfDNA concentration and fragmentation and primary tumor and metastasis location. (**A**) cfDNA fragmentation; and (**B**) cfDNA levels according to primary tumor location; (**C**) cfDNA fragmentation; and (**D**) cfDNA levels according to metastatic location (* *p* < 0.05).

**Figure 7 cancers-12-01754-f007:**
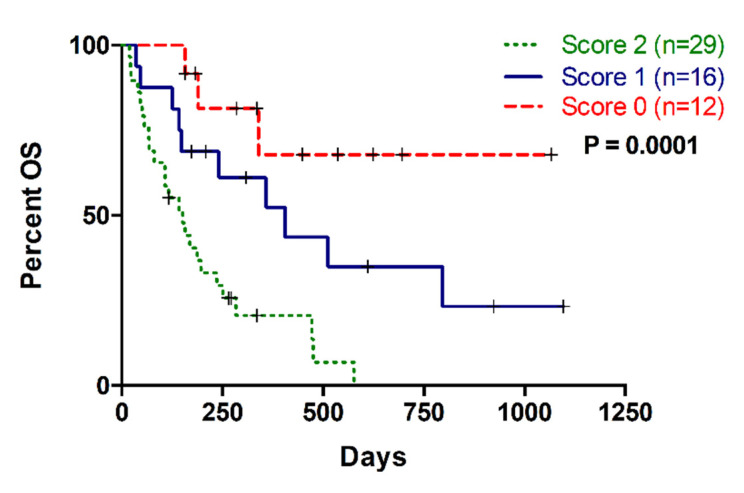
Multi-parametric analysis combining CA19-9 and cfDNA based liquid biopsy markers (MAF, cfDNA concentration and cfDNA fragmentation). Score 0: no positive markers; Score 1: One positive marker; Score 2: More than one positive markers.

**Figure 8 cancers-12-01754-f008:**
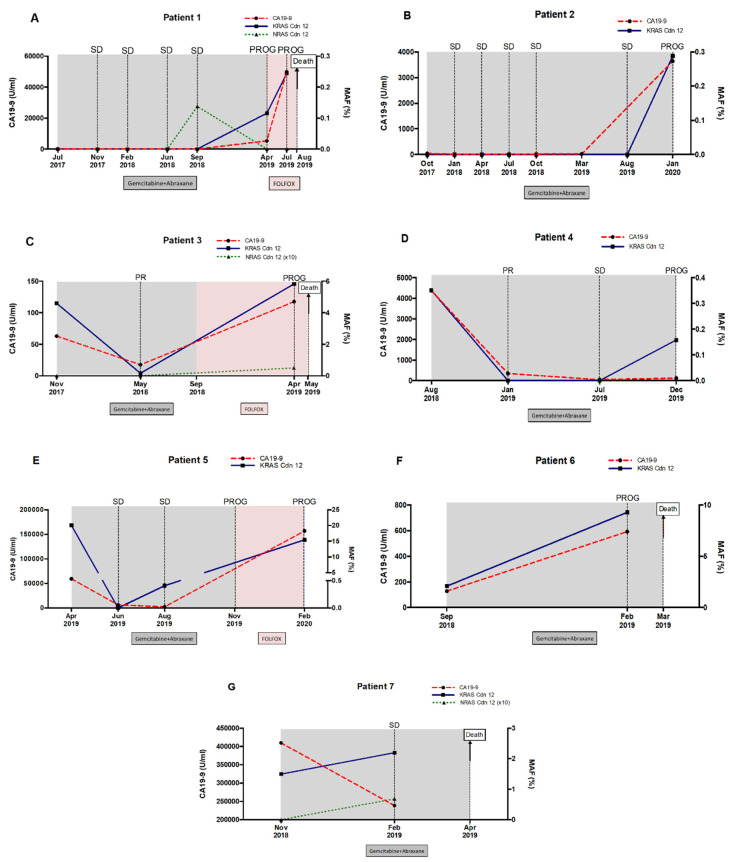
Circulating RAS mutation allele fraction (MAF) enables monitoring of disease progression and response to therapy in metastatic PDAC patients. Circulating MAF (%) was compared to CA19-9 (U/mL), in monitoring response to therapy and disease progression in 8 (**A**–**G**) metastatic PDAC patients.

**Figure 9 cancers-12-01754-f009:**
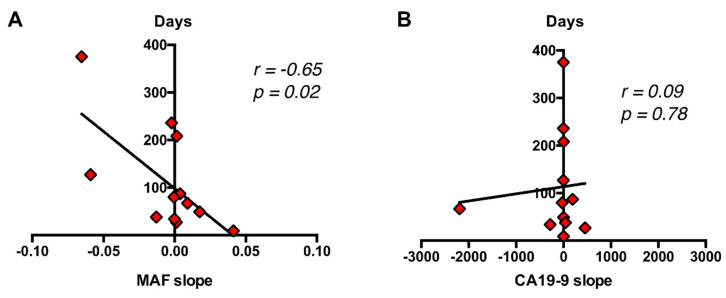
Correlation of dynamics of circulating RAS MAF and CA19-9 with patient’s outcome and survival. (**A**) A significant correlation was found between the increase in MAF (**B**), but not in CA19-9 and survival time.

**Table 1 cancers-12-01754-t001:** Baseline characteristics of patients.

Patient Characteristics		Number of Cases (*n* = 61)
Age (median, range)		65 (40–84)
Sex	Male	34 (55.7%)
	Female	27 (44.3%)
ECOG	0	17 (27.9%)
	1	31 (50.8%)
	2	10 (16.4%)
	3	3 (4.9%)
1st line treatment	Gemcitabine	3 (4.9%)
	Gemcitabine/nab-paclitaxel	39 (63.9%)
	Gemcitabine/nab-paclitaxel/FOLFOX	4 (6.6%)
	FOLFIRINOX	11 (18%)
	No treatment	4 (6.6%)
Survival	Alive	19 (31.1%)
	Dead	42 (68.9%)
Disease progression	Yes	45 (73.8%)
	No	11 (18%)
	Not valuable (No treatment or surgery)	5 (8.2%)
Tissue availability	Yes	43 (70.5%)
	No (Cytology)	18 (29.5%)
Primary tumor location	Tail	17 (27.9%)
	Body	25 (41%)
	Head	18 (29.5%)
	Body-Tail	1 (1.6%)
Number of metastatic lesions	One location	26 (42.6%)
	More than one location	35 (57.4%)
Metastatic lesions location	Hepatic lesions	48 (78.7%)
	Non-hepatic lesions	13 (21.3%)
Tissue Biopsy RAS status ^1^	RAS mutated	33 (76.7%)
	RAS wild-type	10 (23.3%)
Liquid Biopsy RAS status	RAS mutated	47 (77%)
	RAS wild-type	14 (23%)

^1^ For the analysis of RAS mutational status, primary tumor tissue was available in 70.5% (43/61) of patients.

**Table 2 cancers-12-01754-t002:** Overall survival analysis.

Variables	Death Occurrence	Median OS (Days)	HR (95%CI)	*p*
Primary Tumor Location				
Body/Tail	28/42	187	0.818 (0.402–1.667)	*p* = 0.5802
Head	12/17	173.5		
Metastatic Location				
Hepatic	35/48	157	2.403 (1.218–4.738)	*p* = 0.0114
Non-hepatic	6/13	339		
Number of Metastasis				
1	16/26	197.5	0.739 (0.398–1.372)	*p* = 0.3380
≥2	25/35	176		
KRAS mutation status plasma				
MUT	37/47	169	3.455 (1.736–6.876)	*p* = 0.0004
WT	4/14	372.5		
KRAS mutation status tissue				
MUT	24/33	197	2.102 (0.933–4.734)	*p* = 0.0730
WT	5/10	440		
KRAS mutation status plasma (with tissue paired sample)				
MUT	25/32	216.5	3.09 (1.364–6.997)	*p* = 0.0068
WT	4/11	537		
CA19-9				
<45,500 U/mL	32/50	202.5	2.272 (1.407–4.930)	*p* = 0.0408
>45,500 U/mL	8/10	125		
cfDNA concentration				
<26.46 ng/mL	8/21	285	2.468 (1.302–4.681)	*p* = 0.0057
>26.46 ng/mL	31/38	149.5		
MAF				
<0.351%	6/10	310	2.212 (1.099–4.452)	*p* = 0.0261
>0.351%	31/37	142		
cfDNA fragmentation				
<38.08%	28/45	197	2.637 (1.1–6.321)	*p* = 0.0297
>38.08%	11/13	116		

**Table 3 cancers-12-01754-t003:** Progression-free survival analysis.

Variables	Disease Progression	Median PFS (Days)	HR (95%CI)	*p*
Primary Tumor Location				
Body/Tail	33/40	152	0.783 (0.364–1.685)	*p* = 0.5318
Head	10/14	81.5		
Metastatic Location				
Hepatic	35/43	86	2.565 (1.333–4.937)	*p* = 0.0048
Non-hepatic	10/13	272		
Number of Metastasis				
1	19/23	127	0.86 (0.465–1.591)	*p* = 0.6304
≥2	26/33	139		
KRAS mutation status plasma				
MUT	38/42	93.5	3.84 (1.974–7.469)	*p* < 0.0001
WT	7/14	313.5		
KRAS mutation status tissue				
MUT	27/32	122.5	2.495 (1.176–5.294)	*p* = 0.0172
WT	7/9	382		
KRAS mutation status plasma (with tissue paired sample)				
MUT	27/30	142.5	3.41 (1.572–7.395)	*p* = 0.0019
WT	7/11	472		
CA19-9				
<45,500 U/mL	35/45	143	3.013 (1.12–8.103)	*p* = 0.0289
45,500 U/mL	9/10	72		
cfDNA concentration				
26.46 ng/mL	11/20	149.5	2.190 (1.199–4.001)	*p* = 0.0107
>26.46 ng/mL	34/36	86.5		
MAF				
<0.351%	8/9	175	2.015 (0.9834–4.129)	*p* = 0.0556
>0.351%	30/33	85		
cfDNA fragmentation				
<38.08%	33/42	145	3.137 (1.313–7.494)	*p* = 0.0101
>38.08%	12/13	81		

**Table 4 cancers-12-01754-t004:** Multivariate Analysis.

Variables	OS		PFS	
	*p* Value	HR (95%CI)	*p* Value	HR (95%CI)
KRAS mutation status plasma	0.011	5.692 (1.497–21.636)	0.001	8.631 (2.311–32.236)
MAF	0.047	1.070 (1.001–1.143)	0.280	1.035 (0.972–1.103)

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
