# Peer review of "Circulating Cell-Free DNA-Based Liquid Biopsy Markers for the Non-Invasive Prognosis and Monitoring of Metastatic Pancreatic Cancer"

_cancers, 2020, doi:10.3390/cancers12071754_

Round 1

Reviewer 1 Report

The present manuscript presented by Toledano et Al. seems to have a big impact in order to investigate the both clinical and prognostic values in PDAC metastatic patients, using a non-inavsive approach. The concept of the present study seems to valid, in which, the analyses of markers could be offer a weapon to monitor the pathophysiology of PDAC patients. Furthermore, looking for the methodology used in this study, we would like to suggest to the authors, some important changes to increase the value of present work.

We would like to stimulate the scientists to have a look at the following list of comments.

Major Comments:

1) In our personal opinion, the sampling of patients in this study is not completely correct, because of the authors should to include only the paired samples (liquid biopsy and FFPE tissue) for each patient. In fact, the authors, included 61 metastatic PDAC patients (mPDAC), but they were able to obtain Istological confirmation in 43/61 patients enrolled. This aspect is to much important because of the authors should:

a) To confirm the presence of PDAC pathology

b) Nevertheless, they should to evaluate the histopathology of the tumor (i.e. to exclude the presence of other histologies; i.e. IPMN carcinoma).

c)  Indeed, the analyses on FFPE tissue, could highlight the real Kras mutational status for each patient.

In addition, the methodology used to extract DNA from the tissue biopsy seems not be described very well. Probably, laser capture microdissection(LCM) could be the best approach in order to establish the pure ratio of RAS mutated allele. Looking at this approach, the author could match exactly the mutational status of tissue biopsy with respect to liquid biopsy, for each patient.

2) The authors should provide a complete table reporting the paired mutational analyses (tissue vs plasma) for each patient.

3) The big group of patients was treated with GEM+ABRAXANE. How much of them had a paired analyses in tissue and plasma?

4) The authors assumed that all patients enrolled were mPDAC. Are there locally advanced PDAC (LAPDAC) patients?

5) The percentage of KRAS mutation was less that 80%. The authors included all  importants hot spot during genetic analyses, but the frequency of mutation seems to low. How the authors can explain it? How many mutations were found at codon 61 of KRAS?

6) Did the authors used pure PDAC cell line in order to set their system?

Author Response

Responses to reviewer 1:

Question 1.- In our personal opinion, the sampling of patients in this study is not completely correct, because of the authors should to include only the paired samples (liquid biopsy and FFPE tissue) for each patient. In fact, the authors, included 61 metastatic PDAC patients (mPDAC), but they were able to obtain Istological confirmation in 43/61 patients enrolled. This aspect is to much important because of the authors should:

  1. a) To confirm the presence of PDAC pathology.
  2. b) Nevertheless, they should to evaluate the histopathology of the tumor (i.e. to exclude the presence of other histologies; i.e. IPMN carcinoma).
  3. c)  Indeed, the analyses on FFPE tissue, could highlight the real Kras mutational status for each patient.

In addition, the methodology used to extract DNA from the tissue biopsy seems not be described very well. Probably, laser capture microdissection(LCM) could be the best approach in order to establish the pure ratio of RAS mutated allele. Looking at this approach, the author could match exactly the mutational status of tissue biopsy with respect to liquid biopsy, for each patient. LCM

Answer 1.- We thank the reviewer for these comments, which allow us to clarify important points of our study. We would like to emphasize that this study was not designed to assess the concordance between tissue and plasma RAS analyses. For this reason, patients in whom it was not possible to determine the RAS mutational status in tissue, due to limited biopsy material, were also included in our study, thus reflecting the actual clinical situation. However, it is important to note that all patients included in our study had confirmed diagnosis of metastatic PDAC by pathological analysis of tumor tissue (n=43) or cytological analysis (n=18), and by computed tomography. This statement has now been included in the Material and Methods section (page 13, lines 296-298) to further clarify this point. We now also provide more details on the RAS mutation analysis in tissue, including the methodology used to extract DNA from the tissue biopsy. Specifically, the IdyllaTM plattform (Biocartis), that utilizes microfluidics processing with specific cartridges and all reagents on board, was employed for RAS mutation analysis in tissue. The process, including nucleic acid extraction, is fully automated, and results indicating WT KRAS mandate testing for NRAS mutations using another specific cartridge (Page 14, lines 319-324). On the other hand, we agree with the reviewer that exhaustive analyses on FFPE tissue, including LCM, could undoubtedly establish the pure ratio of RAS mutated allele in tissue. However, these types of analyses rely on the availability of biopsy material to be adequately performed, that is not the case for a significant number of PDAC patients, and this is indeed the issue that liquid biopsy may effectively address. This statement has been also included in the Discussion section (page 12, lines 242-246).

Question 2.-The authors should provide a complete table reporting the paired mutational analyses (tissue vs plasma) for each patient.

Answer 2.- We agree with the reviewer that this is relevant information, and we now have included Supplementary Table 1detailing the mutational analysis results for each patient both in tissue and plasma.

Question 3.- The big group of patients was treated with GEM+ABRAXANE. How much of them had a paired analyses in tissue and plasma?

Answer 3.- In our study 74.4% of those patients treated with gemcitabine plus nab-paclitaxel has a paired tissue/plasma RAS analysis. These data are also available in supplementary Table 1.

Question 4.-The authors assumed that all patients enrolled were mPDAC. Are there locally advanced PDAC (LAPDAC) patients?

Answer 4.- As stated above in the response to question 1, all patients included in our study had confirmed diagnosis of metastatic PDAC by computed tomography. No locally advanced PDAC patients were included in this study.

Question 5.- The percentage of KRAS mutation was less that 80%. The authors included all  importants hot spot during genetic analyses, but the frequency of mutation seems to low. How the authors can explain it? How many mutations were found at codon 61 of KRAS?.  

Answer 5.- The percentage of patients with KRAS mutation in our study was comparable to other cohort studies (Kinugasa et al, 2015; Park et al, 2018; Wang et al, 2019). This statement has now been included in the Results section (Page 2, Lines 70-71), and one new citation (Park et al, 2018) have been incorporated into manuscript. On the other hand, KRAS mutation in codon 61 was found in 1 patient (new Supplementary table 1).

Question 6.- Did the authors used pure PDAC cell line in order to set their system?

Answer 6.-. We did not performed studies on pancreatic cancer cell lines.

Reviewer 2 Report

This study investigates the role of cell free DNA as a prognostic biomarker in metastatic pancreatic ductal adenocarcinoma. The authors used BEAMing technique to quantitate RAS mutation allele fraction and TapeStation to assess DNA fragmentation. They analyzed the utility of RAS mutation allele fraction, cfDNA levels and fragmentation in predicting patient's survival and the risk of hepatic metastases. The manuscript deals with a "hot" topic and describes analysis of a relatively large cohort. There are some interesting insights that can be learned from the data presented. However, the statistical analysis and more specifically, the choice of cutoffs for each variable should be better explained.

Major comments

  • The authors chose different cutoff values for each parameter. In some of the parameters a different cutoff was chosen for the evaluation of PFS and OS. For example, the MAF cutoff for OS is 0.351% and the MAF cutoff for PFS is 8.303%. The authors should provide explanations for the choice of cutoffs. These could be based on previous recommendations in the literature. Alternatively, the mean or the median could serve as cutoffs to compare groups.
  • The authors found statistically significant association between DNA fragmentation and tumor localization and hepatic metastasis. The biological sense behind this findings is not clear. Why would you expect higher fragmentation in DNA from pancreatic body/tail tumors? Or tumors that metastasize to the liver? Could this finding be "noise" and not a real biological phenomenon? The authors should discuss the potential biological reasons for this finding. The claim that body/tail tumors are more aggressive than head tumors and that tumor with liver metastasis are more aggressive than tumors with other metastasis should be supported by citations from the literature.
  • The combined scoring system developed by the authors is based on CA19-9 level, MAF, cfDNA concentration and cfDNA fragmentation, and the authors defined score 0 as negative for all markers; score 1 as positive for 1 marker; and score 2 as positive for 2, 3 or 4 markers. It is not clear how the authors define positive or negative values for cfDNA concentration and fragmentation. This information should be provided.

Minor comment

  • In the tables the units for cfDNA concentration are ng/µl. Do the authors mean the concentration of extracted DNA? It would be better if the authors provided the concentration as ng per ml of blood since the final DNA concentration is dependent on the elution volume that can differ between samples and kits.

Author Response

Responses to reviewer 2:

Question 1.- The authors chose different cutoff values for each parameter. In some of the parameters a different cutoff was chosen for the evaluation of PFS and OS. For example, the MAF cutoff for OS is 0.351% and the MAF cutoff for PFS is 8.303%. The authors should provide explanations for the choice of cutoffs. These could be based on previous recommendations in the literature. Alternatively, the mean or the median could serve as cutoffs to compare groups.

Answer 1.- We appreciate the reviewer's comment, and now provide more details on the selection of cutoff values. We made use of the Kaplan Scan (KaplanScan) feature of R2, where an optimum survival cut-off is established based on statistical testing instead of for example just taking the average or median. The Kaplan scanner separates the samples of a dataset into two groups based on values of variable of interest. In the order of values, it uses every increasing value as a cutoff to create 2 groups and test the p-value in a logrank test. This is the explanation of why in some cases a different cutoff was chosen for the evaluation of PFS and OS. This explanation has now been included in the Statistical Analyses sub-section of Materials and Methods, to further clarify this issue (Page 14, lines 332-336).

Question 2.- The authors found statistically significant association between DNA fragmentation and tumor localization and hepatic metastasis. The biological sense behind this findings is not clear. Why would you expect higher fragmentation in DNA from pancreatic body/tail tumors? Or tumors that metastasize to the liver? Could this finding be "noise" and not a real biological phenomenon? The authors should discuss the potential biological reasons for this finding. The claim that body/tail tumors are more aggressive than head tumors and that tumor with liver metastasis are more aggressive than tumors with other metastasis should be supported by citations from the literature.

Answer 2.- We thank the reviewer for these comments, which allow us to clarify this issue. We argue that the results of a recent study showing that KRAS mutations in pancreatic cancer are predominantly carried by short cfDNA fragments (Zvereva et al, 2020) may be the biological explanation for our observation that, in parallel with our KRAS MAF results, a higher cfDNA fragmentation was found in those patients with tumors located in the body/tail of the pancreas or with hepatic metastases versus other metastatic lesions. Also, we now provide additional citations from the literature supporting that body/tail PDAC may have more aggressive tumor biology and higher metastasis rate compared to PDAC in the head which may explain worse clinical outcomes (van Erning et al, 2018, PMID: 30264642; Dreyer et al, 2018; PMID: 29341146; Birnbaum et al, 2019). This statement has now been included in the Discussion section (Page 13, lines 265-270).

Question 3.-  The combined scoring system developed by the authors is based on CA19-9 level, MAF, cfDNA concentration and cfDNA fragmentation, and the authors defined score 0 as negative for all markers; score 1 as positive for 1 marker; and score 2 as positive for 2, 3 or 4 markers. It is not clear how the authors define positive or negative values for cfDNA concentration and fragmentation. This information should be provided.

Answer 3.- In our combined scoring system, positive or negative values were assigned depending on whether the corresponding marker was above (positive) or below (negative) the cut-off with prognostic value in OS. This definition has now been included in the results section (page 9, lines 171-173) to clarify this point.

Question 4 (Minor comment).- In the tables the units for cfDNA concentration are ng/µl. Do the authors mean the concentration of extracted DNA? It would be better if the authors provided the concentration as ng per ml of blood since the final DNA concentration is dependent on the elution volume that can differ between samples and kits.

Answer 4.- We would like to thank the reviewer for this suggestion and following this recommendation cfDNA concentration is now provided as nanograms of cfDNA per milliliter of plasma. Tables 2-3 and figures 5-6 have been modified accordingly.

Round 2

Reviewer 1 Report

The authors performed some additional comments to their manuscript and replied step by step to question of reviewer. Only one point remains still open: the pairing analyses concerning tissues vs liquid biopsies. However the authors explain sufficientely their thesis. We are thinking the the manuscript is ready for the publication. No scientific comments and variation are needed.

Author Response

We would like to thank the reviewer for helpful comments, which have improved the manuscript.

Reviewer 2 Report

The authors provided satisfactory explanation for most of the question, however, no good explanation was provided to the choice of cutoffs in the study.

Broadly speaking, decision regarding cutoff can be data-driven and can take into account the clinical context. The authors determined the cutoff based only on statistical analysis of the data and this could lead to results that are statistically significant but clinically insignificant. Specific example for this problem could be the cutoffs used for mutant variant allele frequency (VAF) for PFS and OS. Using the suggested cutoffs most of the patients would have a VAF value between 0.351% (the OS cutoff) and 8.303%(the PFS cutoff) - all these patients are supposedly with increased  risk of dying from disease (above OS cutoff) and decreased risk for disease progression (below PFS cutoff). The authors should provide a cutoff that has some clinical relevance. To support a cutoff, the sensitivity and specificity of specific cutoff should be provided.

Author Response

Question 1.- The authors provided satisfactory explanation for most of the question, however, no good explanation was provided to the choice of cutoffs in the study.

Broadly speaking, decision regarding cutoff can be data-driven and can take into account the clinical context. The authors determined the cutoff based only on statistical analysis of the data and this could lead to results that are statistically significant but clinically insignificant. Specific example for this problem could be the cutoffs used for mutant variant allele frequency (VAF) for PFS and OS. Using the suggested cutoffs most of the patients would have a VAF value between 0.351% (the OS cutoff) and 8.303%(the PFS cutoff) - all these patients are supposedly with increased  risk of dying from disease (above OS cutoff) and decreased risk for disease progression (below PFS cutoff). The authors should provide a cutoff that has some clinical relevance. To support a cutoff, the sensitivity and specificity of specific cutoff should be provided.

Answer 1.- We agree with the reviewer that this aspect deserves further clarification and we now have improved the choice of cutoffs in our study. In this regard, time-dependent ROC curves can link marker value with clinical needs when the primary endpoint of interest is a time-to-event endpoint such as OS. Therefore, we have now used a SurvivalROC package implemented in R software to find optimal cut-off values in OS analyzing time-dependent ROC curve. The optimal cut-off value was chosen by minimizing the sum of false negative rate and false positive rate. In each case, the cut-off with prognostic value for OS was also tested for prognosis of PFS. When the optimal cut-off chosen with ROC curves was not able to separate statistically the groups according to the Kaplan-Meier analysis (CA19-19 and cfDNA fragmentation), the R2 Genomics Analysis and Visualization Platform (http://r2.amc.nl) was used to find a cut-off value, using the Kaplan Scan (KaplanScan) feature. Using this approach, a new cut-off was identified for cfDNA concentration (26.46 instead of 25.34 ng/ml). We also now provide the sensitivity and specificity for each cutoff. We have included new text in the Material and Method section (page 14, line 333 and lines 336-341) and in the Results section (page 6, lines 107-109; page 7, lines 139-141; page 8, lines 142 and 144-146; page 9, lines 168-170), and tables 2 and 3 as well as figures 3, 5 and 7 have been modified. These changes are tracked in the revised version of the manuscript. These modifications better support the conclusions of our study, and we hope that our responses and the modifications performed are adequate and satisfactory.

Round 3

Reviewer 2 Report

The authors addressed all the comments and elaborated on the statistical methodology. The results are clear and the findings are of interest.

In my opinion the manuscript is of merit and is suitable for publication.